# Ocean fronts as decadal thermostats modulating continental warming hiatus

Mi-Kyung Sung [1] ✉, Soon-Il An [2,3] ✉, Jongsoo Shin [4], Jae-Heung Park [5], Young-Min Yang[6], Hyo-Jeong Kim [7] & Minhee Chang[1]

Over the past decade, an unexpected cooling trend has been observed in East Asia and North America during winter. Climate model simulations suggest that this pattern of stalled warming, besides accelerated warming, will repeat throughout the course of global warming, influenced by the natural decade-long variations in the climate system. However, understanding the exact factors affecting the pace of warming remains a challenge. Here we show that a pause in warming over continental areas—namely, local warming hiatus—can be accompanied by excessive heat accumulation north of the ocean fronts. This oceanic condition, often manifesting in the form of marine heatwaves, constrains the subseasonal growth of atmospheric planetary waves, significantly increasing the likelihood of cold extremes in downstream continents. Our results underscore the importance of closely monitoring changing ocean fronts in response to human-induced warming, which can potentially reshape the inherent decade-long fluctuations within regional climates over the long term.

In July 2023, the global mean temperature (GMT) reached an unprecedented high, coinciding with record-breaking heatwaves and widespread wildfires that swept across the globe[1,2]. Historical data illustrates a significant increase in the frequency, duration, and intensity of heat events over recent decades, aligning with the accelerated upward trajectory of the GMT[3–5]. In contrast, amidst this warming trend, winter temperatures remain unexpectedly cold[6–8] compared to the warming pathways projected in climate models. The persistence of cold extremes has led to a debate about whether wider winter temperature fluctuations are inherent to global warming, directing scientific attention to the rapid decline in Arctic sea ice[9–13].

Understanding the cause of recent winter cold extremes remains an open question, and the contentious debate surrounding the influence of diminished sea ice coverage persists[11–16]. Reduced sea ice amplifies heat transfer from the exposed ocean to the atmosphere, causing anomalous warming in the Arctic. However, climate models only partially replicate the observed continental cooling in the mid-latitudes[14,15,17,18]. This disparity between observations and models has propelled research focus into the internal variability within the climate system pertinent to decadal variations in winter climate[19–21].

A potential key to unraveling the issue of recurrent cold extremes may lie in the intrinsic nature of winter climates conducive to the growth of planetary waves[22], i.e., a chain of continental-scale cyclones and anticyclones. Stronger land-sea thermal contrast, accelerated jet streams that impact orography, and abundant potential energy residing in the steepened north-south mean temperature gradient all fuel growth of planetary waves[23] to concomitantly bring intense cold or

[1]Climate and Environmental Research Institute, Korea Institute of Science and Technology, Seoul, Republic of Korea. [2]Department of Atmospheric Sciences/ Irreversible Climate Change Research Center, Yonsei University, Seoul, Republic of Korea. [3]Division of Environmental Science and Engineering, Pohang University of Science and Technology, Pohang, Republic of Korea. [4]Woods Hole Oceanographic Institution, Woods Hole, MA, USA. [5]School of Earth and Environmental Sciences, Seoul National University, Seoul, Republic of Korea. [6]Key Laboratory of Meteorological Disaster, Ministry of Education (KLME)/Joint International Research Laboratory of Climate and Environment Change (ILCEC)/Collaborative Innovation Center on Forecast and Evaluation of Meteorological Disasters (CIC-FEMD), Nanjing University of Information Science and Technology, Nanjing, China. [7]Low-Carbon and Climate Impact Research Centre, School of Energy and Environment, City University of Hong Kong, Hong Kong SAR, China. ✉e-mail: mksung@kist.re.kr; sian@yonsei.ac.kr

warm advections across the mid-latitudes. The complex interplay of internal climate system factors that modulates the preferred planetary wave responses can lead to divergent decadal trends in regional winter temperatures[24,25], subsequently altering the likelihood of cold events for a given decade. In this sense, an in-depth understanding of the internal climate variability that shapes the behavior of planetary waves can provide valuable insights into the future outlook of regional climate.

In this study, we delve into the underlying physical processes responsible for the decadal variations in cold extremes by leveraging initial-condition large ensemble experiments using a single climate model[26]. This approach offers distinct advantages as the ensemble spread evolves solely through internal variability. Specifically, we conducted 28 ensemble experiments utilizing the Community Earth System Model[27], imposing gradual quadrupling of $CO_2$ level (ramp-up experiment; see Methods). Notably, our model experiments exhibit a remarkable agreement with the observed features.

## Recurrent local hiatus in global warming

Regional surface air temperatures respond to global warming depending on geographical location and topographic conditions. This is especially the case during winter, as can be seen in Fig. 1. Conspicuous cooling trends are found in East Asia and North America, contrasting with steeper warming trends in Europe and the Arctic, particularly in recent decades (Supplementary Fig. 1a).

We probe the cause of these distinct decadal cooling trends, which appear to manifest simultaneously in the two distant continents, by tracing the temporal changes (Fig. 1b, c). Wide fluctuations appear between winters (black line); however, their running averages exhibit a prominent warming trend before 1993 in East Asia (blue curve) and before 2002 in North America (red curve). Afterward, the trends shift toward cooling, with a 10-year time-lag between the two regions. Compared to the annual-mean GMT (bars), whose upward trend changes to a slowdown in the 2000s−the so-called global warming hiatus period[28,29]−, the flip of trend in East Asia tends to precede global-scale changes, whereas that of North America roughly coincides with the GMT trend.

The GMT has recently resumed its upward trajectory, but it is uncertain if the cooling trends in the two continental areas will return to warming in the coming decade. We seek insight into the possible drivers of these long-term temperature fluctuations using future

projections for the two concerned areas simulated through ramp-up ensemble experiments (Fig. 2). For easier comparison with the observation, we present only the running-averaged time series, as shown by the colored lines in Fig. 1. Clear upward trends are found in all ensemble experiments (bright-colored lines) as the atmospheric $CO_2$ concentration increases 1% annually until it quadruples by 2140. During warming, decadal temperatures largely fluctuate, as is clear in the sampled ensemble simulation (darker colored lines). This example time series shows recurring decade-long cooling periods, the onset of which is marked by inverted triangles−we hereafter refer to such cooling periods as a local warming hiatus.

Occurrences of the local warming hiatus in 28 ensemble experiments are displayed in Fig. 2b. The shortest box marks a cooling trend that persisted for approximately 10 years, and the longest one shows that the trend could persist for as long as 20 years. Totals of 34 and 37 local hiatus events were respectively captured in East Asia and North America by current thresholds (Methods), illustrating that the regional warming trend will repeatedly stall and resume as global warming progresses. The recurrence of paused warming periods inherently stems from decadal variability in winter temperatures, manifesting as unusually deep cooling trends that counteract the overall forced warming trend. This suggests that recently observed cooling in winter temperatures aligns with the intrinsic aspect of the climate system.

The irregular occurrence of decadal cooling periods in the two distant regions (Fig. 2b; green color) suggests independent drivers specific to each region. Nonetheless, a comparison of the local trends with GMT (bars; Fig. 2a) tells the relationship between regional and global temperatures; in the sampled simulation, North America passes through an accelerated warming period in the 2060 s (see red curve surpassing the dotted line), while the GMT is under a relaxation phase (bright-colored bars), where the GMT is lower than the ensemble average. In this example simulation, the global-scale changes seem to better coincide with the changes in East Asia, unlike the observation, both increasing anomalously in the 2040 s and cooling in the 2060 s. However, these coincidences are not consistent in other ensemble simulations (Supplementary Fig. 1c). The results show that the warming rate in GMT only limitedly affects the regional continental temperatures during winter. Unlike winter temperatures, which fluctuate widely at a decadal time scale, summer regional climates continue an upward trend, exhibiting only minor decadal fluctuations[24,25] (Supplementary Fig. 1).

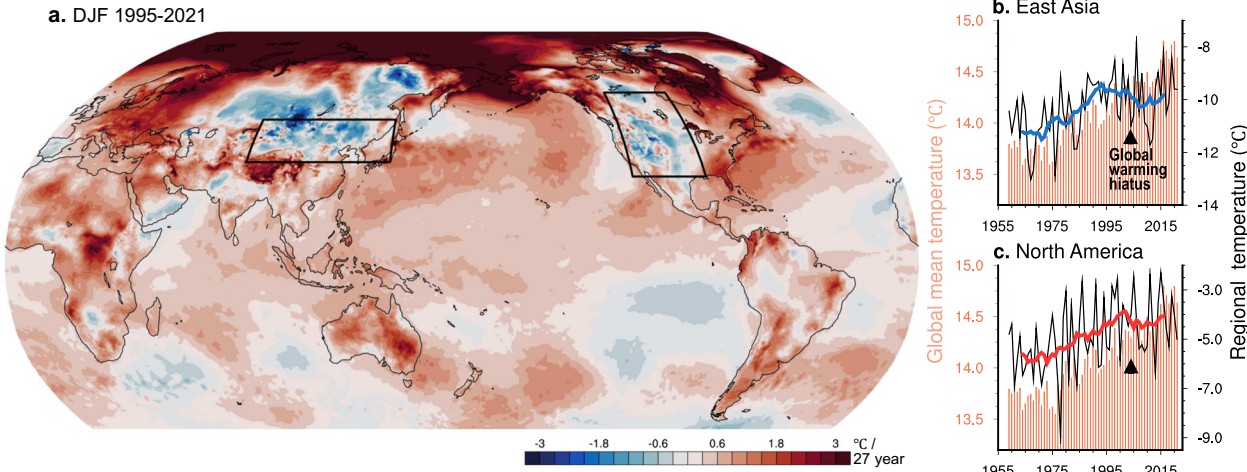

**Fig. 1 | Observed temperature trends during the winters of 1995/96-2021/22.**
**a** Global map of linear trends in winter-mean (Dec-Feb) temperatures. **b**, **c**. Temporal variations of winter-mean regional temperatures in **b** East Asia and **c** North America, area-averaged for the boxed regions in **a**. Black lines denote winter-mean temperatures, and their averages within 11-yr running windows are presented in blue and red curves. Orange bars denote the annual-mean global mean temperature (GMT).

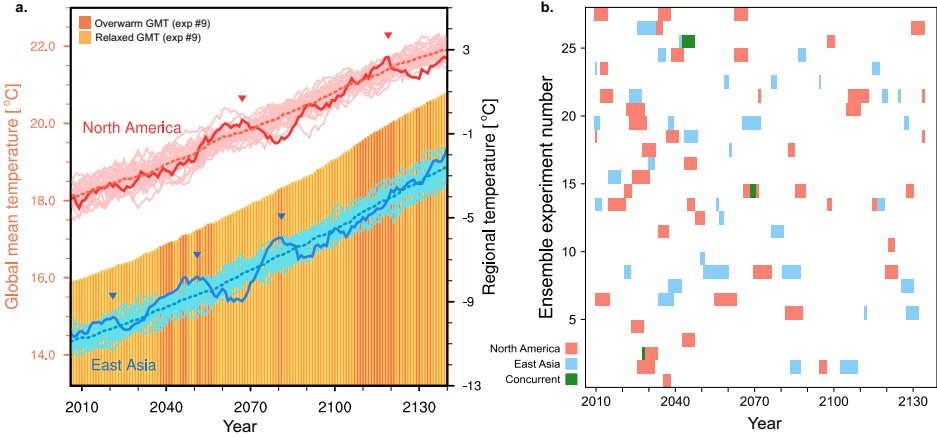

**Fig. 2 | Decadal temperature changes in ramp-up and occurrence of local warming hiatuses. a** An 11-year running mean time series of winter temperatures over North America (red) and East Asia (blue) in ramp-up experiments. Bright-colored lines represent each individual ensemble simulation in which the ninth is highlighted in darker colors. Dotted lines denote the ensemble average, and upside-down triangles denote the onset of a local warming hiatus. Bars depict the annual-mean GMT (ninth ensemble member result), highlighted by a darker color when warmer than the ensemble averages. **b** Filled boxes denote periods when regional temperatures show a steep negative trend within a running window (<−0.5 °C/11 years; Methods). Vertical axis represents 28 ensemble experiments.

## Ocean fronts as decadal thermostat

Now we examine the processes that could drive regional winter temperatures anomalously low over a decade. Here, the term anomalous in the model simulations denotes a deviation from the average of all ensemble experiments at a given time such that we can define an anomalously cold period irrespective of the global warming trend (Methods). We composited all hiatus events detected in the ramp-up experiments to determine their time evolution from the early beginning till the mature phase. As a hiatus period tends to be preceded by an overwarming period, we separately compare the antecedent warm and ensuing cold periods. Setting an onset year as lag 0 (i.e., the years marked by upside-down triangles in Fig. 2a), a decade from lag−4 to lag +5 years is assigned to the overwarm period and the following 10 years (lag +6 ~ +15 years) to the mature hiatus period.

This classification well captures overly warm or cold East Asian (blue-red shading; Fig. 3a) and North American (Supplementary Fig. 2) temperatures. Compared to the observed reference (Fig. 3b), the model well portrays causative atmospheric circulations (contours); an anticyclonic circulation over warm East Asia accompanied with a cyclonic subpolar branch in the west[30]. In a cold decade, nearly symmetrical features are found, though with opposing signs, in the simulations and observations, indicating that essentially the same processes but with opposite phases are involved in the decadal warming and cooling. These decadal averaged atmospheric circulations highly resemble a typical atmospheric pattern that leads to warm and cold spells in East Asia within a winter season (Supplementary Fig. 3a)[31]. The structural similarity between subseasonal and decadal scales is also observed in the results for North America (Supplementary Fig. 4). The similarity between the vastly different time scales could stem from repeated manifestations of opposing phases of subseasonal planetary waves during a cold or warm decade. Corroborating this, during mature hiatus periods, daily temperatures show an increased frequency of severe cold extremes (< − 2σ) rather than an increase in mild cold events (Supplementary Fig. 5).

The cause of a local warming hiatus is presumed to be related to the origin of a recurring planetary wave, which provides a favorable background flow condition. Figure 3 shows that attention should be directed to the North Atlantic, which has historically shown a close association with East Asian cold surges[31,32]. A cyclonic circulation forms over Greenland during a warm decade in East Asia and a contrasting anticyclonic circulation is observed during a cold decade. These atmospheric patterns, changing signs over decades, arise along with underlying oceanic changes (red boxed region) to cold sea surface temperature (SST; purple-brown shading) near Greenland during an overwarm period of East Asia and a contrasting warm SST during a mature hiatus period. These changes are summarized in Fig. 3c, d (shading; upper panel), illustrating a pronounced shift from cold to warm SST.

Slow variation in ocean temperature suggests that a coupled atmosphere-ocean process in the Atlantic basin[33] may be responsible for the decadal variation in East Asia by modulating atmospheric planetary waves. Regarding the physical process, changes near the Gulf Stream is worthy of note. In this narrow ocean front region, where the SST sharply decreases northward (Supplementary Fig. 6), the steep SST gradient acts to sustain an atmospheric thermal gradient through differential heat release. As a result, the atmospheric storm track that develops to stir the maximum gradient region becomes anchored along the ocean front (Supplementary Fig. 6)[34,35].

Along with the SST changes before and after the hiatus onset, the meridional gradient over the Gulf Stream shifts from anomalously strong to weak conditions (brown curve in Fig. 3c, d; Methods; also see Supplementary Fig. 6). Under the condition of a steeper oceanic gradient, the atmospheric temperature gradient gets readily restored against the relaxing effect of the storm track[35,36]. In contrast, a smoothed oceanic gradient retards the restoration of the atmospheric thermal gradient, making the air-sea coupling less efficient[37]. Accordingly, storm track activity (blue-red shading; lower panel of Fig. 3c, d) intensifies in the presence of a strengthened oceanic gradient and then weakens as the oceanic gradient declines. The model simulations and observations consistently demonstrate coincidence in these changes in the North Atlantic with temperature shifts in downstream East Asia (gray curve; bottom panel). Though the observational SST gradient seems to slightly precede East Asian temperatures, the North Atlantic storm track activity exhibits better coincidence with downstream temperatures.

Understanding how gradual changes in the North Atlantic storm track influence downstream winter climate can be enhanced through a comparison with subseasonal changes. Notably, the spatial changes in storm track during warming hiatus periods in East Asia closely resemble the temporary suppression of the storm track typically observed within a winter season, especially during cold snaps in East Asia (Supplementary Fig. 3b). In terms of magnitude, the alterations in

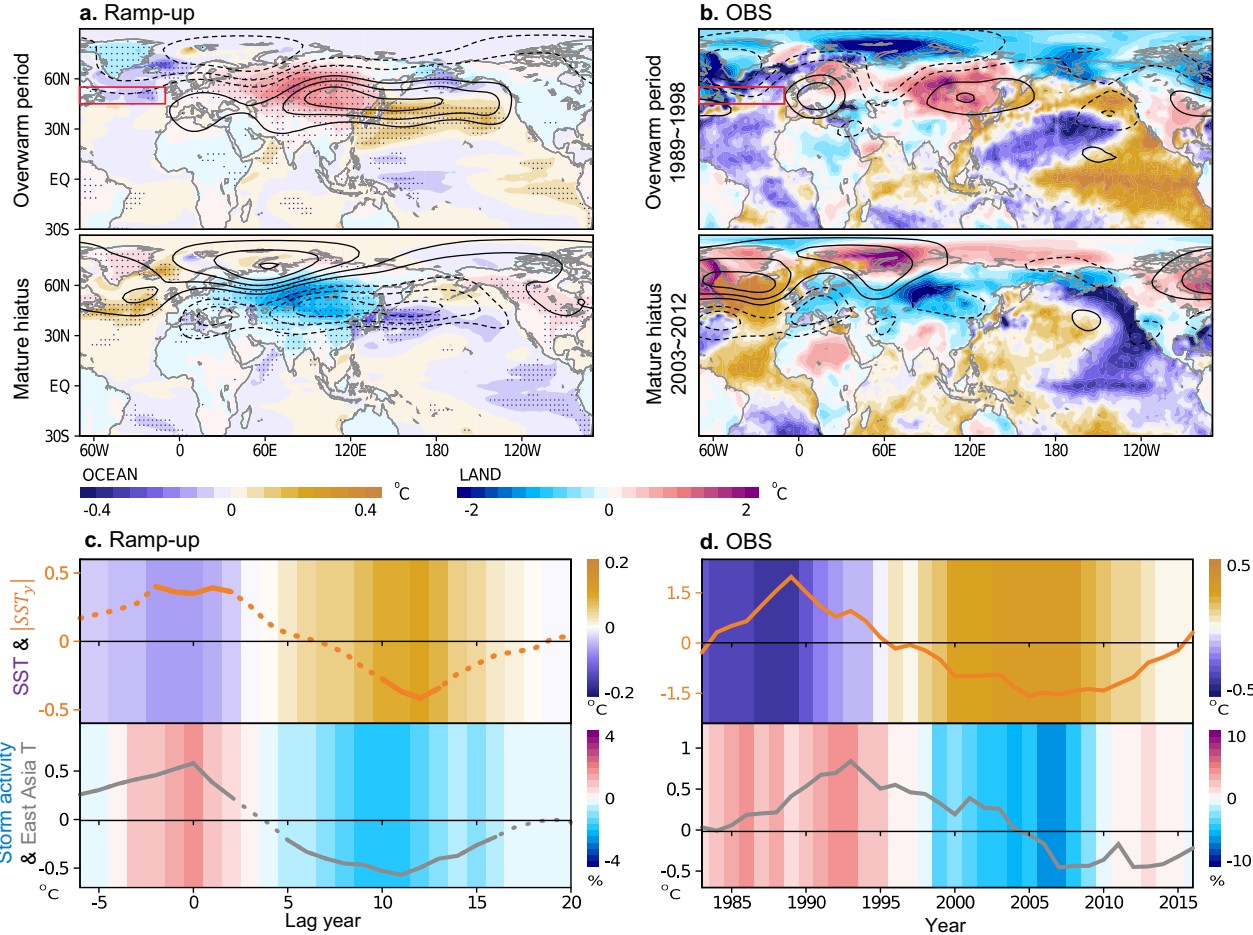

**Fig. 3 | Atmosphere and ocean conditions during the overwarm and mature hiatus decades of East Asia.** Winter temperatures over land and ocean (shading; surface air temperature and sea surface temperature (SST)) during the overwarm (upper) and mature hiatus (lower) periods in **a** ramp-up experiments and **b** observations. Contours denote the average atmospheric circulation in the upper troposphere (geopotential height anomaly at 300 hPa) during each decade, in which solid (dashed) lines represent anticyclonic (cyclonic) circulations (5 gpm interval for model output and 15 gpm interval for observation without a zero line).

**c**, **d** (Upper) Time evolution of SST (11-year running averaged; shading) near Greenland (red boxed region in **a** and **b**; 70–10°W, 45–55°N) and north-south SST gradient (|SST$_y$|) near the Gulf Stream (brown curve; non-dimensionalized; see Methods and Supplementary Fig. 6). (Lower) Relative change in storm track activity anchored over the Gulf Stream (shading; Methods) and East Asian temperature (gray curve). Stippling in **a** and the solid curve in **c** indicate values significant at the 95% confidence level. Data in the left column are derived from ramp-up experiments and the right column presents an observational reference.

storm track activity due to the oceanic constraint changes are much weaker compared to subseasonal fluctuations. Nevertheless, the consistency in spatial patterns between the different time scales suggest that the gradual variations in storm track activity can influence the likelihood of a particular subseasonal planetary wave response.

A simple idealized model experiment was conducted to demonstrate the impact of storm track suppression on the downstream atmosphere. In this experiment, we introduced vorticity forcing[38,39], specifically, an anomalous divergence of vorticity flux (Supplementary Fig. 3c, d; see Methods). This forcing pattern can become more frequent over a decade when the background storm track activity is anomalously weakened. The model produced a planetary wave response characterized by an anticyclonic branch over subarctic Siberia and a cyclonic branch downstream. As mentioned earlier, such a planetary wave pattern establishes conducive conditions for triggering cold surge outbreaks in East Asia[31]. Conversely, intensified storm track activity due to a steepened ocean frontal gradient over the North Atlantic can increase the probability of a planetary wave response that favors a warm spell in East Asia, consequently leading to a decade-long warming.

In the real atmosphere, more complex processes such as diabatic heating in conjunction with vorticity forcing could be involved in

generating a planetary wave response[40–42], as demonstrated by more sophisticated climate model experiments in earlier studies[33,43,44]. To further verify the role of the regional SST gradient in the Gulf Stream in affecting temperatures far downstream, we also carried out another set of atmospheric model experiments in which the north of the Gulf Stream was forced with a strong oceanic warming pattern derived from a reference period (Warm-Gulf experiment; see Methods). The average response of winter temperatures in 20 ensemble experiments shows a pronounced cooling in Siberia and northern East Asia (Supplementary Fig. 7b). Although this forced simulation does not allow for atmospheric feedback to the ocean, the result supports that long-term variability in the North Atlantic frontal region are capable of modulating a decadal temperature shift in East Asia.

Likewise, overwarm and hiatus periods in North America coincide with changes near the North Pacific ocean front, the Kuroshio Extension (Supplementary Figs. 2 and 6). In the same manner as the Gulf Stream affecting East Asian temperatures, long-term changes in the Kuroshio Extension appear to influence the downstream climate[45–47]. However, in the case of the Pacific basin, atmosphere-ocean coupled processes may be more important than in the Atlantic, as explored below.

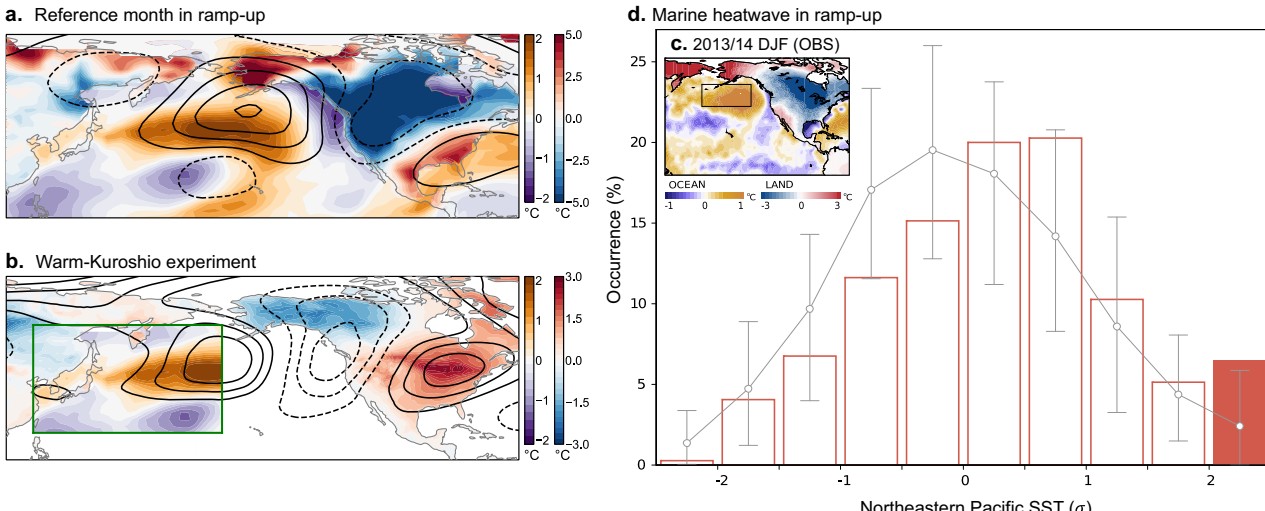

**Fig. 4 | Influences of weakened North Pacific front on continental cold wave and marine heatwave. a** Temperatures over the ocean (purple-brown shading) and land (blue-red shading) during a reference period, chosen to force the Warm-Kuroshio experiment (Methods). Contours denote the corresponding atmospheric circulations (300 hPa geopotential height anomaly drawn at ±50, ±150, ±250 gpm) **b** Average response of planetary wave and temperature in the Warm-Kuroshio experiments (contours with 10 gpm interval). Values present deviations from the

control experiment (Methods). The green box indicates the region where the anomalous SST forcing was imposed. **c** Anomalous oceanic warming and continental cooling during the winter of 2013/14. **d** Probability of a marine heatwave over the eastern North Pacific (boxed area in **c**) during a mature hiatus of North America in ramp-up experiments (bar). The filled bar indicates a significant deviation from non-hiatus periods (contour), whose 5th and 95th percentiles are presented via error bars (Methods).

## Marine footprint of local warming hiatus

A warming hiatus accompanied with the oceanic change is a decadal phenomenon, but as illustrated in Fig. 4a, monthly fields frequently exhibit concurrent temperature patterns, i.e., anomalous warming in the northern flank of the Kuroshio Extension alongside freezing temperatures in North America. To investigate the downstream influence of oceanic changes along the Kuroshio Extension, a set of Warm-Kuroshio experiment was conducted, similar to the Warm-Gulf experiment (Methods). In this experiment, the reference SST depicted in Fig. 4a was applied to the western North Pacific region (boxed in Fig. 4b) to focus on the influence of the ocean front. Despite the restricted regional forcing, the average atmospheric response reveals a planetary wave pattern similar to that shown in Fig. 4a[47], while the resulting cold temperatures in North America appear narrower and shifted northwestward compared to the reference data.

In spite of some differences, this experimental result suggests important implications regarding the atmosphere-ocean interaction over the North Pacific that may be critical in leading to decade-long cooling in North America. To explore this issue in more detail, we shift our focus to the eastern North Pacific, downstream of the Kuroshio Extension, where recurring winter marine heatwaves (or Pacific warm blob) have recently attracted attention[48,49]. Figure 4c shows a warm blob event that occurred in the winter of 2013/14 and triggered heavy scientific and media attention due to its unprecedented intensity and persistence[48], with SST nearly 2 °C above normal across thousands of kilometers that lasted through 2015. This event left devastating damage to fishery and marine environments from toxic algal blooms[49,50] and was followed by another warm blob event in 2019-2020[51].

Ramp-up experiments propose that successive emergence of warm blob events is an inherent feature of a warming hiatus in North America, as can be seen in Fig. 4d. During this period, the maximum probability of the SST in the warm blob region (boxed in Fig. 4c) shifts to the right compared to normal conditions (i.e., non-hiatus period; Methods). Eventually, the probability of exceptional warm events (> +2σ) significantly increases (filled bar). The result shows that a devastating marine heatwave in the eastern North Pacific, surpassing the severity anticipated solely from the global warming

trend, is inherently coupled to a seemingly opposing freezing winter in North America, featured by an unusually frequent occurrence of cold spells.

This counter-intuitive relationship can be explained from the physical mechanism behind the 2013/14 warm blob case, which was promoted by enhanced oceanic warm advection from upstream, in addition to reduced heat loss to the atmosphere[48,52]. A positive SST-cloud feedback also proved to be an important contributor that amplified a moderately warm SST anomaly into a warm blob[49]. Certainly, the warmer oceanic temperature in the northern flank of the Kuroshio Extension can provide a positive background for the development of a warm blob event, allowing anomalous heat transport downstream. The weakened storm track activity passing over the weak ocean front region further aids in the formation of a marine heatwave by reducing upward surface heat flux and impeding vertical mixing. Moreover, an anticyclonic wave response over the North Pacific, shown in the Warm-Kuroshio experiment, intensifies the SST warming by activating these feedback processes[52]. Consequently, the presence of the upstream oceanic warming enhances the likelihood of marine heatwaves off the west coast of North America through intensified atmosphere-ocean interactions.

It should be noted here that the warm blob is not only the result of feedback, but also serves as a cause of exaggerated continental cold[53], as the warm blob condition further suppresses storm track activity. A simple model experiment demonstrates that a weakened storm track activity deepens a cyclonic planetary wave branch over North America, thereby exacerbating the cold wave (Supplementary Fig. 4c, d). This means that marine heatwaves, which are anticipated to become more prevalent as global warming intensifies[54], may be even more serious in the eastern North Pacific during a decade when the warming trend in North America is stagnated.

## Implications for future change

The role of ocean front as a modulator of decadal climate raises questions regarding the predictability of ocean front variability and its response to global warming. Although the observed features in the North Atlantic and North Pacific basins appear similar, the underlying processes driving decadal changes differ. The thermal gradient

structure and intensity near the Gulf Stream are strongly influenced by the Atlantic Meridional Overturning Circulation (AMOC), whose local surface branch constitutes the Gulf Stream[55,56]. In the North Pacific, remote influences from the tropical Pacific, such as the impact of ENSO transmitted through planetary wave propagation, play a significant role[57,58]. However, in addition to these basin-scale circulations, a multitude of processes operating across various timescales contribute to shaping decadal variability. These include oceanic waves affecting mid-latitude gyre circulation, multi-year persistence in the ocean mixed layer, and heat exchange with the atmospheric disturbances[57,59,60]. As a result, the observed decadal cooling in the two continental regions can be largely attributed to the combined effects of internal climate system factors that become integrated into the memory of the mid-latitude ocean mixed layer. As the decadal variability in ocean fronts exhibits inherent red-noise aspects, it imposes limitations on predicting mid-latitude winter climates on a decadal scale.

However, this does not imply that the observed winter cooling trends are solely attributable to natural climate variability. A comparison between the ramp-up experiment and the pre-industrial (PI) control simulation, which represents climate variability without anthropogenic forcing, reveals that anthropogenic warming acts to amplify decadal variability in East Asia but attenuates it in North America (Fig. 5a; red-blue colors). Consequently, in East Asia, decade-long cooling events become more frequent in the presence of anthropogenic forcing (Fig. 5b), while they become fewer in North America (Fig. 5c).

Divergent changes in the two ocean basins during global warming[61] (Fig. 5a; purple-brown colors) may account for the inconsistent responses on the continents. In general, climate models predict a weakening of the AMOC under global warming[55,56], which leads to a cooling trend in the subarctic North Atlantic. Consequently, the mean oceanic gradient near the Gulf Stream strengthens, and the westerly jet stream aloft extends further eastward[62]. These environmental changes enhance the coupling between the storm track and oceanic variability, while also displacing the downstream planetary wave branch eastward[63,64], favoring increased decadal variability in Eastern Eurasia as illustrated in Fig. 5a. If the observed slowdown of the AMOC indeed reflects a response to anthropogenic forcing, the recent cooling trend in East Asia may, to some extent, be attributed to this forced response.

On the other hand, the subarctic North Pacific experiences a steeper warming trend due to ocean stabilization, which leads to a shallower ocean mixed layer[65]. The subarctic SST warming condition over the North Pacific can contribute to freezing winters in North America by weakening frontal gradient and storm track activity. However, the decline in oceanic heat capacity and memory under global warming[65,66] ultimately dampen decadal climate variability in North America, reducing chances of decade-long cold trend in the long run. These complex facets of global warming make it challenging to assess the influence of anthropogenic forcing on recent winter cooling trends.

Nonetheless, it is worth noting that climate models generally simulate smoother oceanic frontal gradients (Supplementary Fig. 6) and weaker decadal variability compared to observations[56,60,67]. This discrepancy in climate model representation leads to a reduced level of atmosphere-ocean coupling over ocean frontal regions compared to what is observed in reality, presumably contributing to a significant underestimation of the internal decadal variability of atmospheric circulations[68–70]. Given this context, local warming pauses, which inherently reflect decadal variability within the climate system, may be more frequent in the real world than projected in warming scenarios. Improving our understanding of the climatic influence of ocean fronts holds the key to gaining valuable insights into regional climate change.

## Methods
### Observational data
The observational surface air temperature, upper tropospheric circulation (300 hPa geopotential height), and storm track activity shown in this study are based on ERA5 data[71] starting from 1959. HadISST v1.1 was used to present the SST, which has $1° × 1°$ horizontal resolutions[72]. Observational analysis focused on the winter season (December to the following February) up to the winter of 2021/22; all observational variables were detrended before analyses except for those in Fig. 1. For SST, a linear trend was estimated for the 1900-2021 base period to minimize the influence of interdecadal variability.

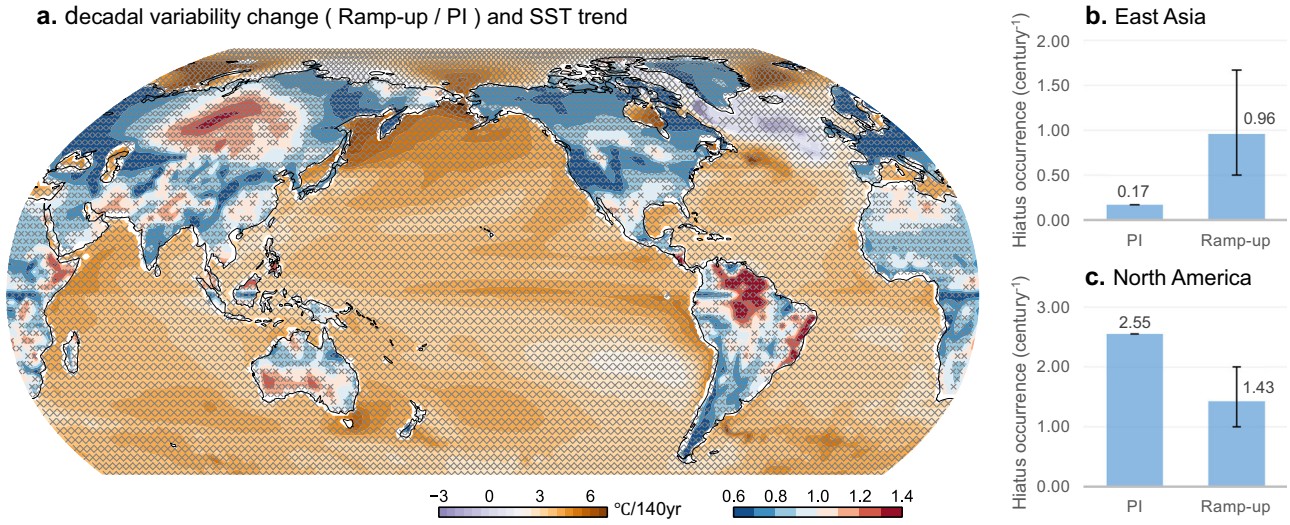

**Fig. 5 | Influence of anthropogenic forcing on local warming hiatus. a** Relative change in decadal variability of surface air temperature over land (red-blue shading; variance ratio of 11-year running mean winter temperatures in ramp-up experiments and pre-industrial (PI) control simulation). Values greater than one indicate increased decadal variability and smaller values indicate decreased variability under anthropogenic warming. Purple-brown shading indicates SST trends in the ramp-up experiment. Regions with statistically significant changes at the 95% confidence level are marked by small crosses. Probability of a warming hiatus (number of events per century) over (**b**) East Asia and (**c**) North America in the ramp-up and PI experiments. Error bars represent 5th and 95th percentile ranges from ramp-up to compare with PI (Methods).

## Ramp-up and PI experiments

We used the fully coupled Community Earth System Model (CESM) version 1.2.2[27], whose atmosphere and land model components have a horizontal resolution of ~1° with 30 vertical levels. Detailed model configurations can be found in An et al.[73]. The ocean components have a 1° longitudinal and ~0.33° meridional resolutions near the equator that gradually change to 0.5° near the poles with 60 vertical levels. We performed ramp-up experiments by increasing the $CO_2$ concentration 1% per year from 2001 ($1 \times CO_2$, 367 ppm) until it quadrupled in 2140 (1468 ppm). To obtain initial conditions for 28 ensemble simulations, the model was integrated for 900 years under a constant $1 \times CO_2$ level. Initial conditions were then taken from this pre-simulation. Atmospheric variables and the SST were analyzed after reformatting to $2.5° \times 2.5°$ and $1° \times 1°$ horizontal resolutions, respectively, as in the observation. An anomaly of variables in the ensemble experiments was defined as a departure from the ensemble mean smoothed by an 11-year running average. PI control simulation was performed under a constant $CO_2$ level of 284.7 ppm (1850 concentration) for 600 years.

## Warm-Gulf/Kuroshio experiments

To verify the impact of ocean fronts on downstream climates, we conducted two types of experiments using the Community Atmospheric Model version 5 (CAM5), the atmospheric component of CESM. In these experiments, we imposed anomalous SST boundary condition over either the Gulf Stream (20–80°N, 70–10°W) or Kuroshio Extension (20–60°N, 120°E–170°W) regions, while maintaining climatological SST values elsewhere. The anomalous SST patterns, which feature an anomalously weak ocean frontal gradient, were sampled from the daily SST field of the ramp-up experiments. We chose to use daily SST instead of composite as a reference, as the atmospheric response to mid-latitude ocean conditions is sensitive to the steepness of the frontal gradient[69], whereas composite SST tends to smooth out such variations. The integrations were conducted from November 1st to February 28th using 20 different initial conditions under the radiative forcing fixed at the levels of the year 2000. To assess the impact of the oceanic boundary condition, we compared the results of the Warm-Gulf/Kuroshio experiments with the control experiments, with the exception that climatological SST values were prescribed globally. Figure 4b and Supplementary Fig. 7b present the differences between the forced and control experiments. For comparison, the monthly anomalies from the ramp-up experiment for the corresponding date of the sampled SST are presented in Fig. 4a and Supplementary Fig. 7a.

## Stationary wave model experiments

A planetary wave response to storm track change was tested using a stationary wave model, a nonlinear baroclinic model with a dry dynamical core and 14 vertical levels[39]. Idealized point-wise transient eddy vorticity flux divergence/convergence forcing was applied to the winter climatological background flow. Vorticity forcing was computed from observational and ramp-up experimental data as $-\nabla \cdot (\overline{V' \xi'})$, where $\xi$ is the vorticity, and $V$ is the horizontal wind at 200 hPa[38]. Prime denotes an 8-day high-pass filtered wind and vorticity, and the bar represents the monthly or seasonal mean. In the experiment, we examined the mean atmospheric response integrated for 30 days. Unlike the observational anomaly, where the forced planetary wave response and the associated anomalous background condition are superimposed, this simple model output represents a forced wave response alone.

## Local warming hiatus definition

An 11-year running average time series of winter regional surface air temperatures was employed to detect a local warming hiatus in the ramp-up experiments. A temporary trend of the regional temperature time series was assessed in a 11-year running window; when a slope was

steeper than −0.5 °C, the nearest warmest year was defined as the hiatus onset. The −0.5 °C thresholds represent approximately the 5th percentile of the regional temperature trend distributions in both East Asia (80–140°E, 35–50°N) and North America (120–90°W, 30–60°N). The next hiatus event was identified only after the temperature trend recovered above the threshold. Applying this definition to the regional temperatures yielded 34 and 37 hiatus events for East Asia and North America, respectively, in 28 ensemble ramp-up experiments. This number of hiatus events varied depending on the threshold, but the overall results were consistent for slight changes in threshold and window size. The significance of the composite results of the over-warm and mature hiatus period, shown in Fig. 3, was determined by Welch's $t$ test[74] to compare the means of independent samples with unequal variance.

In the PI experiment, which has no long-term warming trend, a warming hiatus event was defined by adopting a stricter cooling trend threshold, namely, −1.13 °C/11 years for East Asia and −1.03 °C/11 years for North America, given the ensemble mean warming trends in the ramp-up experiments, which are +0.63 °C/11 years for East Asia and +0.53 °C/11 years for North America. We note that the overall results are consistent when the same criteria are used for the two regions. For a fairer comparison of ramp-up and PI, in Fig. 5b, c, we calculated a hiatus event probability of ramp-up experiments using the same threshold as PI after removing the ensemble mean trend from the original data.

## SST gradient and storm track

A meridional gradient near the ocean front was computed from the 11-year running average of the total SST at all grid points, and then, an area-average was applied upon a deviation from the climatological mean gradient for the observation and from the ensemble mean gradient for the ramp-up experiments in the vicinity of the Gulf Stream region (40–45°N, 70–30°W in observation; 37–47°N, 70–10°W in model) and the Kuroshio Extension (35–42°N, 140°E–170°W in observation and 38–43°N, 140°E–170°W in model). The different settings of the observation and model are due to differences in climatology (see red boxes in Supplementary Fig. 6). The resulting time series was standardized and composited according to hiatus onset. Storm track activity was defined as the standard deviation of an 8-day high-pass filtered daily geopotential height at 500 hPa[75] for a winter. Then, its decadal variation, obtained with 11-year running averages, was measured at 25°W over 35–55°N range of the North Atlantic and at 155°W over 35–45°N range of the North Pacific (see red dotted lines in Supplementary Fig. 6). Storm track activity displayed in Fig. 3 and Supplementary Fig. 2 was given in percentage units relative to climatology.

## Statistics

Warm blob event probability over the eastern North Pacific was calculated based on the winter-mean SST anomaly averaged over the 180°–140°W, 40–55°N domain. The probability distribution for a mature hiatus period was compared with that of non-hiatus periods, which denote the same year range as the hiatus period in other ensemble experiments. The nonsignificant difference range in Fig. 4d was estimated from a two-sided bootstrap resampling to test against the null hypothesis that H0: $C_O = C_m$, where $C_O$ is the winter-mean eastern North Pacific SST distribution for a randomly selected decade, and $C_m$ is the SST distribution for the same year range in other ensembles. We created 5000 subsets that comprised 37 random single decades (i.e., the number of warming hiatus events for North America in ramp-up experiments) and presented their 5th and 95th percentiles via error bars.

The significant difference ranges between the ramp-up and PI experiments regarding the frequency of local warming hiatus were estimated by creating 5000 subsets that comprised randomly resampled 600-year data (100 years × 6 times) from 28 ramp-up ensembles. Their 5th and 95th percentiles were presented via error bars in Fig. 5b, c.

## Data availability

The data used in this study is available from https://doi.org/10.6084/m9.figshare.22225936.v1[76], ERA5 datasets are publicly available from the European Center for Medium-Range Weather Forecasts (https://climate.copernicus.eu/climate-reanalysis), and HadISST data is publicly available from Met Office Hadley Center (https://www.metoffice.gov.uk/hadobs/hadisst/).

## Code availability

The codes used in this study are available from the corresponding author on request.

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

## Acknowledgements

This work was supported by the National Research Foundation of Korea (NRF) grant funded by the South Korean government (MSIT) (NRF-2018R1A5A1024958 and NRF-2021R1A2C1003934). M.K.S. and M.C. were partially supported by 2022M3K3A1094114. J.H.P. was supported by NRF-2023R1A2C1004083 and Y.M.Y. was supported by NRF-2022R1A2C1013296.

## Author contributions

M.K.S. conceived the idea and designed the study. J.S. performed model experiments and M.C. assisted analyzing data. M.K.S. wrote the first draft and revised it reflecting feedbacks from S.I.A., J.H.P., Y.M.Y. and H.J.K.

## Competing interests

The authors declare no competing interests.
