## [Peer Review File · Nature Communications]

Ocean fronts as decadal thermostats modulating continental warming hiatusEditorial Note: This manuscript has been previously reviewed at another journal that is not operating a transparent peer review scheme. This document only contains reviewer comments and rebuttal letters for versions considered at *Nature Communications*.

REVIEWER COMMENTS

Reviewer #1 (Remarks to the Author):

Disclosure:

Note that this reviewer was not part of the earlier round(s) of the review process. Therefore, at the stage, I will forgo providing another overview of the manuscript and instead attempt to only focus on new comments and questions that were generally not raised by the previous reviewers.

Summary:

While I see that the authors have now conducted additional analysis to support their proposed mechanisms for the importance of oceanic fronts on midlatitude decadal temperature variability, I find the manuscript still to be slightly difficult to interpret. I think this could be addressed by reformatting some of the text to provide a more storyline approach to the main message of the paper. I was particularly confused in the beginning of the study where it still made it sound like the novel contribution of this work was identifying decadal variability in midlatitude wintertime temperatures. As the authors/reviewers point out in the previous revision round, this is not the case. There have been countless studies on identifying the role of internal climate variability on interannual to multi-decadal timescales. Though I see the authors now reference Deser et al. (2014) at L41, to me, this is still an insufficient overview on the previous work assessing the drivers of East Asia and North America temperature variability. More references are needed here, especially related to the work of leveraging single model initial-condition large ensembles (like from the multi-model large ensemble archive). The introduction and abstract of the paper should make it clearer that the mechanism through oceanic fronts is the novel contribution of this work - not the identification of decade-long cooling periods nor changes in the variance of wintertime temperatures associated with external forcing.

Overall, this is an interesting study that will certainly contribute to our understanding of decadal climate variability and change. However, I had to read the study several times to understand the findings and choice of analysis. I encourage the authors to implement major revisions to the text, with a focus on providing ample discussion and context to prior work on evaluating East Asia and North America temperature trend drivers and explaining all of their climate model perturbation experiments for completeness. I have some more specific comments and questions listed below.

Specific Points:

1. L1; I am not sure the title is very intuitive, as it is not necessarily common to refer to internal climate variability and local scales using the word "hiatus." Further, this study only really analyzes two regions – East Asia and North America.
2. L15-17; I understand what the authors were trying to say here, but it is not completely valid. While the global mean surface temperature (GMST) did rise after the early 2000s, it has since entered another brief "hiatus" with a nearly flat trend over the 2016 to 2022 period.
3. L16-17; Is this actually a question in the literature? For example, the National Academy of Science report on attribution found high confidence that global warming will reduce the frequency of cold spells.

4. L17-19; This is not a new finding.
5. L23-25; This is a very strong concluding statement, which I am not sure is necessarily supported given that primarily only one model is used here for the analysis (CESM1).
6. L28-29; This sentence is confusing to me. Why would regional summer heatwaves in 2022 indicate the end of a GMST hiatus?
7. L30-31; But summertime heatwaves can also be unpredictable, especially at S2S timescales? I suggest revising the use of the word prediction here.
8. L31-33; I understand that the authors do not need to cite every paper, but there are some major references missed here (e.g., Cohen et al. 2020; Overland et al. 2021).
9. L39; "...across [the] mid-latitudes."
10. L41-42; Again, I am confused by the use of the word prediction here – especially in reference to climate models (they are not decadal prediction systems).
11. L42-43; But due to chaos theory, how could "precise" winter weather outlooks ever be made at these timescales?
12. L47-48; Are these 28 ensemble members from one global climate model (GCM) or 28 different GCMs? Revise the wording here.
13. L53; Why start in 1995/1996? Based on graph 1b?
14. L87-88; This is not surprising and is well simulated by climate model large ensembles.
15. L112-115; I am not sure this statement is completely supported by Figure 2b. Where does it demonstrate that the decadal cooling is not related to Arctic climate change? There have been dozens of studies on this topic relating Arctic amplification and Arctic sea-ice loss to Siberian/East Asian cooling.
16. L117-118; In my view, this is where this study's main new contributions to the literature begin.
17. L184-189; I find these experiments to be quite interesting. It might be useful to focus more text here in the revision (and less on the previous identifying local hiatus findings (as listed above)).
18. L233-235; Can you be more specific here for the word choice of "devastating" and "freezing"?
19. L314-315; What do you mean by the subarctic warming promoting freezing conditions?
20. L322-323; How does this finding compare to other studies?
21. L464-465; Are 28 ensemble members enough to sample internal variability in boreal winter for these locations? For example, a recent paper found that 100s of ensemble members may be needed to understand Arctic-midlatitude climate linkages.
22. L465; What are "pre-runs"?

Reviewer #2 (Remarks to the Author):

In this study, Sung et al. analyzed and compared that the warming hiatus in East Asia and in North America during the past decades, and found a difference in timing between them. Also, the authors found that the coherence of these hiatus with the SST anomaly in the North Atlantic and in the western North Pacific, individually. These SST anomalies correspond to the SST front intensification in the Gulf Stream and the Kuroshio Extension. Furthermore, the authors compared the projected hiatus in the future based on CMIP6 experiments, and found an increase likelihood of a decade-long cooling in East

Asia, but a decrease in North America. The authors used the connection of SST-front with atmospheric storm track, and with atmospheric wave to account for the impact of SST front future change on the projected hiatus change.

Overall this study is of importance for understanding the future temperature change under the global warming. The finding that hiatus is well coherent to the SST front is vital. But the proposed mechanism that the SST front change in the future causes the hiatus change is not persuasive. Particularly, the authors inferred the processes for the interannual connection between the SST, atmospheric stormtrack, and stationary atmospheric wave anomaly to the decadal timescale. This lacks of evidences. Also, the AGCM experiments and the idealized transient forcing experiments cannot support their argument due to the interaction between multiple timescales (synoptic, seasonal-interannual, and decadal), although the experiments are useful for a seasonal-interannual question study.

More,

The authors attributed the hiatus over the East Asia to the SST front in the Gulf Stream. One previous study (Lei et al, 2021) suggests that the SST warmth in the Kuroshio Extension also plays important role. The mechanism for how the SST front influences the stormtrack has not been revealed clearly in the manuscript.

The authors used the filtered daily synoptic Z300 variance to calculate the stormtrack, but used idealized transient vorticity forcing to figure out the atmosphere wave response to anomalous stormtrack. How to give the transient vorticity forcing based on the transient Z300 variance has not been described clearly.

Reference

Lei, T., S. Li, Luo F, N. Liu, 2020. Two dominant factors governing the decadal cooling anomalies in winter in East China during the global hiatus period. *Int. J. Climatol.*, 40(2), 750-768

Referee #1:

We are deeply grateful to the reviewer for valuable feedbacks, which helped us to clarify the novelty of our work. Below are our responses to the reviewer's comments. Reviewer 1's comments are indicated in black and our responses are indicated in blue.

While I see that the authors have now conducted additional analysis to support their proposed mechanisms for the importance of oceanic fronts on midlatitude decadal temperature variability, I find the manuscript still to be slightly difficult to interpret. I think this could be addressed by reformatting some of the text to provide a more storyline approach to the main message of the paper. I was particularly confused in the beginning of the study where it still made it sound like the novel contribution of this work was identifying decadal variability in midlatitude wintertime temperatures. As the authors/reviewers point out in the previous revision round, this is not the case. There have been countless studies on identifying the role of internal climate variability on interannual to multi-decadal timescales. Though I see the authors now reference Deser et al. (2014) at L41, to me, this is still an insufficient overview on the previous work assessing the drivers of East Asia and North America temperature variability. More references are needed here, especially related to the work of leveraging single model initial-condition large ensembles (like from the multi-model large ensemble archive). The introduction and abstract of the paper should make it clearer that the mechanism through oceanic fronts is the novel contribution of this work - not the identification of decade-long cooling periods nor changes in the variance of wintertime temperatures associated with external forcing.

We appreciate the reviewer for this important comment to make our key findings clearer. Agreeing with the reviewer, we completely reframed the abstract and introduction parts in the revised manuscript. Please see our detailed response below.

Overall, this is an interesting study that will certainly contribute to our understanding of decadal climate variability and change. However, I had to read the study several times to understand the findings and choice of analysis. I encourage the authors to implement major revisions to the text, with a focus on providing ample discussion and context to prior work on evaluating East Asia and North America temperature trend drivers and explaining all of their climate model perturbation experiments for completeness. I have some more specific comments and questions listed below.

In light of the reviewer's comments, we have revised the abstract to clarify the main question of our study, which is about the exact factors influencing the regional warming pace. The abstract now reads:

“Over the past decade, an unexpected cooling trend has been observed in East Asia and North America during winter. Climate model simulations suggest that this pattern of stalled warming, besides accelerated warming, will repeat throughout the course of global warming, influenced by the natural decade-long variations in the climate system. However, understanding the exact factors affecting the pace of warming remains a challenge. Here we show that a pause in warming over continental areas—namely, local warming hiatus—can be accompanied by excessive heat accumulation north of the ocean fronts. This oceanic condition, often manifesting in the form of marine heatwaves, constrains the subseasonal growth of

atmospheric planetary waves, significantly increasing the likelihood of cold extremes in downstream continents. Our results underscore the importance of closely monitoring changing ocean fronts in response to human-induced warming, which can potentially reshape the inherent decade-long fluctuations within regional climates over the long term.”

In addition, the revised introduction part now addresses the following three points:

1. Representative perspectives to explain mid-latitude cooling in previous studies (mainly, the influence of Arctic sea ice reduction) and limitations of climate modelling
2. The aspect of decadal variability of winter temperature as an internal climate system variability
3. Advantages of single model large ensemble approach to resolve the role of internal variability

More specific responses to the reviewer’s comments are given below.

Specific Points:

1. L1; I am not sure the title is very intuitive, as it is not necessarily common to refer to internal climate variability and local scales using the word “hiatus.” Further, this study only really analyzes two regions – East Asia and North America.

When decadal variation is superimposed on forced warming trend, it appears as hiatus trend as shown in Fig. 2. This trend would be such that we will observe more common in the observation.

We changed the title as “Ocean fronts as decadal thermostats modulating continental warming hiatus” to better represent the main finding of our study by highlighting the role of ocean fronts.

As for the term “hiatus”, we would like to keep using it, because

(1) the term describes what we observe directly from the raw observations without statistical detrending (intuitively easy to understand)

(2) such a concept offers methodological advantages. Although a “local warming hiatus” inherently stems from natural decadal climate variability, it indicates an unusually deep cooling period that offsets the forced warming trend. This approach helps to focus on strong cooling events.

(3) we expect the same approach can be applied to other regions, although we have only analyzed two regions here. This approach will provide useful insights for understanding regional climate change.

2. L15-17; I understand what the authors were trying to say here, but it is not completely valid. While the global mean surface temperature (GMST) did rise after the early 2000s, it has since entered another brief “hiatus” with a nearly flat trend over the 2016 to 2022 period.

As we aforementioned, the abstract was now rewritten referring to the role of natural decadal variability regarding the recent cooling trend in the observation.

3. L16-17; Is this actually a question in the literature? For example, the National Academy of Science report on attribution found high confidence that global warming will reduce the frequency of cold spells.

In the revised abstract, the main question of this study is clarified as follows:

“However, understanding the exact factors affecting the pace of warming remains a challenge.”

4. L17-19; This is not a new finding.

New finding is now given as follows:

“Here we show that a pause in warming over continental areas—namely, local warming hiatus—can be accompanied by excessive heat accumulation north of the ocean fronts.”

5. L23-25; This is a very strong concluding statement, which I am not sure is necessarily supported given that primarily only one model is used here for the analysis (CESM1).

We understand the reviewer concern and revised the concluding statement to more carefully deliver the key message as:

“Our results underscore the importance of closely monitoring changing ocean fronts in response to human-induced warming, which can potentially reshape the inherent decade-long fluctuations within regional climates over the long term.”

6. L28-29; This sentence is confusing to me. Why would regional summer heatwaves in 2022 indicate the end of a GMST hiatus?

The introduction part was largely rewritten given the comments of the reviewer. Now it begins as:

“In July 2023, the global mean temperature (GMT) reached an unprecedented high, coinciding with record-breaking heatwaves and widespread wildfires that swept across the globe^{1,2}. Historical data illustrates a significant increase in the frequency, duration, and intensity of heat events over recent decades, aligning with the accelerated upward trajectory of the GMT³⁻⁵. In contrast, amidst this warming trend, winter temperatures remain unexpectedly cold⁶⁻⁸ compared to the warming pathways projected in climate models. The persistence of cold extremes has led to a debate about whether wider winter temperature fluctuations are inherent to global warming, directing scientific attention to the rapid decline in Arctic sea ice⁹⁻¹³.”

7. L30-31; But summertime heatwaves can also be unpredictable, especially at S2S timescales? I suggest revising the use of the word prediction here.

We replaced the expression “unpredictably” with “unexpectedly” in the sentence.

8. L31-33; I understand that the authors do not need to cite every paper, but there are some major references missed here (e.g., Cohen et al. 2020; Overland et al. 2021).

Thank you for recommending those important references. Those studies are now included in the reference list.

9. L39; "...across [the] mid-latitudes."

Revised.

10. L41-42; Again, I am confused by the use of the word prediction here – especially in reference to climate models (they are not decadal prediction systems).

The phrase has been revised to fit a different context:

"The complex interplay of internal climate system factors that modulates the preferred planetary wave responses can lead to divergent decadal trends in regional winter temperatures^{24,25}, subsequently altering the likelihood of cold events for a given decade."

11. L42-43; But due to chaos theory, how could "precise" winter weather outlooks ever be made at these timescales?

That part was revised as:

"In this sense, an in-depth understanding of the internal climate variability that shapes the behavior of planetary waves can provide valuable insights into the future outlook of regional climate."

12. L47-48; Are these 28 ensemble members from one global climate model (GCM) or 28 different GCMs? Revise the wording here.

Taking this comment together with the reviewer's other major comment about the advantage of single model initial-condition large ensembles, we revised the manuscript as follows:

"In this study, we delve into the underlying physical processes responsible for the decadal variations in cold extremes by leveraging initial-condition large ensemble experiments using a single climate model²⁶. This approach offers distinct advantages as the ensemble spread evolves solely through internal variability. Specifically, we conducted 28 ensemble experiments utilizing the Community Earth System Model²⁷, imposing gradual quadrupling of CO₂ level (ramp-up experiment; see Methods)."

13. L53; Why start in 1995/1996? Based on graph 1b?

We chose the period to show the cooling trends in the two distant continents at the same time. Such a context was elaborated as:

"We probe the cause of these distinct decadal cooling trends, which appear to manifest simultaneously in the two distant continents, by tracing the temporal changes in two continental areas (Fig. 1b,c)."

14. L87-88; This is not surprising and is well simulated by climate model large ensembles.

We understand the reviewer's concern and tried to refine the overall context so as not to highlight the finding of decade-long cooling periods. If contextually necessary, we elaborated the aspects of natural decadal variability.

In L87-88 (now in L95-101), we elaborate this context as:

"Totals of 34 and 37 local hiatus events were respectively captured in East Asia and North America by current thresholds (Methods), illustrating that the regional warming trend will repeatedly stall and resume as global warming progresses. The recurrence of paused warming periods inherently stems from decadal variability in winter temperatures, manifesting as unusually deep cooling trends that counteract the overall forced warming trend. This suggests that recently observed cooling in winter temperatures aligns with the intrinsic aspect of the climate system."

15. L112-115; I am not sure this statement is completely supported by Figure 2b. Where does it demonstrate that the decadal cooling is not related to Arctic climate change? There have been dozens of studies on this topic relating Arctic amplification and Arctic sea-ice loss to Siberian/East Asian cooling.

We agree with the reviewer that more rigorous examination would be required to ensure such a statement. That notion has now been removed.

16. L117-118; In my view, this is where this study's main new contributions to the literature begin.

We thank the reviewer for explicitly pointing this out. We have revised the title of our paper in light of this comment of the reviewer as aforementioned.

17. L184-189; I find these experiments to be quite interesting. It might be useful to focus more text here in the revision (and less on the previous identifying local hiatus findings (as listed above)).

Taking this comment into account, together with the reviewer's other comments, we have revised the main text as follows:

(1) The first two subsections of the main text have been merged into one. Revising the manuscript, we have softened the finding of local warming hiatus and clarified the aspect as a result of decadal variability. However, it was difficult to significantly reduce the size of this section, as it demonstrates that the model realistically portrays such temporal variability. This was also contextually necessary in view of the introductory section, which now mentions the discrepancy between observations and models in simulating mid-latitude cooling in response to Arctic sea ice loss.

(2) We elucidated the simple model experiment further in detail in the revised manuscript (L206-215):

"A simple idealized model experiment was conducted to demonstrate the impact of storm track suppression on the downstream atmosphere. In this experiment, we introduced vorticity forcing^{38,39}, specifically, an anomalous divergence of vorticity flux (Extended Data Fig. 3c,d;

see Methods). This forcing pattern can become more frequent over a decade when the background storm track activity is anomalously weakened. The model produced a planetary wave response characterized by an anticyclonic branch over subarctic Siberia and a cyclonic branch downstream. As mentioned earlier, such a planetary wave pattern establishes conducive conditions for triggering cold surge outbreaks in East Asia³¹. Conversely, intensified storm track activity due to a steepened ocean frontal gradient over the North Atlantic can increase the probability of a planetary wave response that favors a warm spell in East Asia, consequently leading to a decade-long warming.”

18. L233-235; Can you be more specific here for the word choice of “devastating” and “freezing”?

We elaborated the sentence to specify the meaning of “devastating” and “freezing”:

“The result shows that a devastating marine heatwave in the eastern North Pacific, surpassing the severity anticipated solely from the global warming trend, is inherently coupled to a seemingly opposing freezing winter in North America, featured by an unusually frequent occurrence of cold spells.”

19. L314-315; What do you mean by the subarctic warming promoting freezing conditions?

The meaning is clarified now as:

“The subarctic SST warming condition over the North Pacific can contribute to freezing winters in North America by weakening frontal gradient and storm track activity.”

20. L322-323; How does this finding compare to other studies?

In previous studies, it was found that climate models tend to inadequately capture air-sea interaction over oceanic frontal regions compared to real world. We complemented the implications of this shortcoming in climate models regarding the probability of local warming hiatus as follows:

“Nonetheless, it is worth noting that climate models generally simulate smoother oceanic frontal gradients (Extended Data Fig. 6) and weaker decadal variability compared to observations^{56,60,67}. This discrepancy in climate model representation leads to a reduced level of atmosphere-ocean coupling over ocean frontal regions compared to what is observed in reality, presumably contributing to a significant underestimation of the internal decadal variability of atmospheric circulations⁶⁸⁻⁷⁰. Given this context, local warming pauses, which inherently reflect decadal variability within the climate system, may be more frequent in the real world than projected in warming scenarios.”

21 L464-465; Are 28 ensemble members enough to sample internal variability in boreal winter for these locations? For example, a recent paper found that 100s of ensemble members may be needed to understand Arctic-midlatitude climate linkages.

It appears that the reviewer is referencing Peings et al. (2021). They concluded, based on “single-year time slice experiments” of PAMIP, that 100 ensemble members may be still influenced by internal variability. In our study, we integrated a total of 28 ensemble members

over a 140-year period each. From these, we selected 34 and 37 composite cases to examine the influence of ocean fronts on the atmosphere. Considering each case spans a 10-year period, our results are derived from an ensemble size equivalent to 340 and 370 years, respectively. Although our work focuses on decadal relationships, the actual physical process driving the remote linkage between ocean fronts and downstream climate operates on a seasonal time scale. In this context, our sample size seems sufficient to draw reasonable conclusions.

22. L465; What are "pre-runs"?

That confusing wording was elaborated as:

"To obtain initial conditions for 28 ensemble simulations, the model was integrated for 900 years under a constant 1×CO₂ level. Initial conditions were then taken from this pre-simulation."

Reviewer #2:

We appreciate the reviewer for the valuable comments, which have helped us to improve the clarity of our work. Below are our responses to the reviewer's comments. Reviewer 2's comments are indicated in black and our responses are indicated in blue.

In this study, Sung et al. analyzed and compared that the warming hiatus in East Asia and in North America during the past decades, and found a difference in timing between them. Also, the authors found that the coherence of these hiatus with the SST anomaly in the North Atlantic and in the western North Pacific, individually. These SST anomalies correspond to the SST front intensification in the Gulf Stream and the Kuroshio Extension. Furthermore, the authors compared the projected hiatus in the future based on CMIP6 experiments, and found an increase likelihood of a decade-long cooling in East Asia, but a decrease in North America. The authors used the connection of SST-front with atmospheric storm track, and with atmospheric wave to account for the impact of SST front future change on the projected hiatus change.

Thanks for evaluating our work. From this comment, we realized that our description of the model experiments was not sufficient. In this study, we analyzed 28 initial condition ensembles integrated in a single model (CESM) and clarified this point in the revised manuscript introduction part (L53-58) as follows:

"In this study, we delve into the underlying physical processes responsible for the decadal variations in cold extremes by leveraging initial-condition large ensemble experiments using a single climate model²⁶. This approach offers distinct advantages as the ensemble spread evolves solely through internal variability. Specifically, we conducted 28 ensemble experiments utilizing the Community Earth System Model²⁷, imposing gradual quadrupling of CO₂ level (ramp-up experiment; see Methods)."

Overall this study is of importance for understanding the future temperature change under the global warming. The finding that hiatus is well coherent to the SST front is vital. But the proposed mechanism that the SST front change in the future causes the hiatus change is not persuasive. Particularly, the authors inferred the processes for the interannual connection between the SST, atmospheric stormtrack, and stationary atmospheric wave anomaly to the decadal timescale. This lacks of evidences. Also, the AGCM experiments and the idealized transient forcing experiments cannot support their argument due to the interaction between multiple timescales (synoptic, seasonal-interannual, and decadal), although the experiments are useful for a seasonal-interannual question study.

Thank you for bringing attention to the time scale gap, which was not clearly conveyed in the original manuscript. As highlighted by the reviewer, the linkage between ocean front regions and downstream continents operates across various timescales. In our initial manuscript, we briefly explained the cause of remote linkage at a decadal time scale, attributing it to altered frequency of subseasonal planetary wave response that brings cold extremes (Extended Data Fig. 5). However, we recognized the necessity for a more thorough explanation to clarify our points, regarding the different time scales.

In the revised manuscript, we have made two enhancements:

(1) We explicitly provide a comparison of the features of the storm track and transient vorticity forcing during hiatus decades and subseasonal cold spells, as illustrated in Extended Data Fig. 3 and 4 (which are also presented below).

(2) In the main text (L197-215), we provide a detailed explanation of how the persistently weakened or strengthened storm track conditions, influenced by oceanic constraints, can impact the likelihood of subseasonal planetary wave responses.

This newly included description is as follows:

“Understanding how gradual changes in the North Atlantic storm track influence downstream winter climate can be enhanced through a comparison with subseasonal changes. Notably, the spatial changes in storm track during warming hiatus periods in East Asia closely resemble the temporary suppression of the storm track typically observed within a winter season, especially during cold snaps in East Asia (Extended Data Fig. 3b). In terms of magnitude, the alterations in storm track activity due to the oceanic constraint changes are much weaker compared to subseasonal fluctuations. Nevertheless, the consistency in spatial patterns between the different time scales suggest that the gradual variations in storm track activity can influence the likelihood of a particular subseasonal planetary wave response.

A simple idealized model experiment was conducted to demonstrate the impact of storm track suppression on the downstream atmosphere. In this experiment, we introduced vorticity forcing^{38,39}, specifically, an anomalous divergence of vorticity flux (Extended Data Fig. 3c,d; see Methods). This forcing pattern can become more frequent over a decade when the background storm track activity is anomalously weakened. The model produced a planetary wave response characterized by an anticyclonic branch over subarctic Siberia and a cyclonic branch downstream. As mentioned earlier, such a planetary wave pattern establishes conducive conditions for triggering cold surge outbreaks in East Asia³¹. Conversely, intensified storm track activity due to a steepened ocean frontal gradient over the North Atlantic can increase the probability of a planetary wave response that favors a warm spell in East Asia, consequently leading to a decade-long warming.”

Extended Data Figure 3. Remote forcing for an East Asian cold period. **a, b.** Anomalous conditions related to an East Asian warming hiatus (upper; ramp-up) and winter cold spell (lower; observation) in terms of (a) geopotential height at 300 hPa (contour; gpm unit) and surface air temperature (shading) and (b) storm track change (shading; relative intensity (%) to climatology denoted by contour). Unlike the anomalies associated with warming hiatus obtained from ramp-up experiments, the anomalies associated with cold spell were computed via linear regression onto observational monthly temperatures over East Asia. **c.** Transient eddy vorticity forcing (see methods) associated with East Asian warming hiatus (left) and cold spell (right). **d.** Stationary wave response (contour; streamfunction) to point-wise divergent transient eddy vorticity forcing (shading) from the stationary wave model (Methods). Unlike the anomaly pattern shown in (a), anomalous background flow conditions over the North Atlantic driving the planetary wave response were not superimposed in this simple model output.

Extended Data Figure 4. Remote forcing for a North America cold period. **a, b.** Anomalous conditions related to a North American warming hiatus (upper; ramp-up) and winter cold spell (lower; observation) in terms of (a) geopotential height at 300 hPa (contour; gpm unit) and surface air temperature (shading) and (b) storm track change (shading; relative intensity (%) to climatology denoted by contour). Unlike the anomalies associated with warming hiatus obtained from ramp-up experiments, the anomalies associated with cold spell were computed via linear regression onto observational monthly temperatures over North America. **c.** Transient eddy vorticity forcing (see methods) associated with North America warming hiatus (left) and cold spell (right). **d.** Stationary wave response (contour; streamfunction) to point-wise divergent transient eddy vorticity forcing (shading) from the stationary wave model (Methods). Unlike the anomaly pattern shown in (a), anomalous background flow conditions over the North Pacific driving the planetary wave response were not superimposed in this simple model output.

These complementary results, demonstrating the consistency of atmospheric variabilities over different time scales, support the validity of AGCM and idealized transient forcing experiments.

More,

The authors attributed the hiatus over the East Asia to the SST front in the Gulf Stream. One previous study (Lei et al, 2020) suggests that the SST warmth in the Kuroshio Extension also plays important role.

Thanks for recommending an interesting study. Lei et al. (2020) proposed two factors influencing East Asia winter cooling; i.e., Arctic sea ice reduction and SST cooling in the Kuroshio extension. Their study demonstrates a positive correlation between Eastern China temperature and local SST, indicating that cold SST condition might be related with cold temperature in China. However, when interpreting this positive correlation to determine

causality, we may need to be cautious, because their study also finds that (in their figure 4a) a notable portion of this positive correlation comes from atmospheric forcing that cools the ocean.

Given this complex context, rather than highlighting the local influence of the Kuroshio Extension, we decided to focus on their modelling work regarding the role of Arctic sea ice. Lei et al. (2020) is now referenced in the introduction section of our revised manuscript.

The mechanism for how the SST front influences the storm track has not been revealed clearly in the manuscript.

The influence of ocean fronts on storm track primarily manifests through turbulent heat flux that acts to sustain atmospheric baroclinicity against the relaxing effect by storm track (Sampe et al. 2010, Nakamura et al. 2008, Small et al. 2014, etc.). This relation can also be seen in Fig. A below, which shows prominent positive correlation over the ocean front regions, indicating that the ocean drives the atmosphere in those regions. When comparing overwarm and mature hiatus periods, the results show stronger positive correlations during overwarm periods (see lower panels), suggesting favorable conditions for storm track development in the presence of steeper ocean frontal gradient.

Referring to these studies, we elaborated as to how decadal variations in the ocean front affect storm track in L185-192.

“Along with the SST changes before and after the hiatus onset, the meridional gradient over the Gulf Stream shifts from anomalously strong to weak conditions (brown curve in Fig. 3c,d; Methods; also see Extended Data Fig. 6). Under the condition of a steeper oceanic gradient, the atmospheric temperature gradient gets readily restored against the relaxing effect of the storm track^{35,36}. By contrast, a smoothed oceanic gradient retards the restoration of the atmospheric thermal gradient, making the air-sea coupling less efficient³⁷. Accordingly, storm track activity (blue-purple shading; lower panel of Fig. 3c,d) intensifies in the presence of a strengthened oceanic gradient and then weakens as the oceanic gradient declines.”

Fig. A. Correlation coefficients between monthly SST and turbulent heat flux at each grid point during overwarm (upper), mature hiatus (middle) periods and their difference (lower) for East Asia (left; Gulf Stream) and North America (right; Kuroshio Extension).

The authors used the filtered daily synoptic Z300 variance to calculate the stormtrack, but used idealized transient vorticity forcing to figure out the atmosphere wave response to anomalous stormtrack. How to give the transient vorticity forcing based on the transient Z300 variance has not been described clearly.

The way we calculated the transient vorticity forcing is now explained in the Methods section (L569-572) as:

“Vorticity forcing was computed from observational and ramp-up experimental data as $-\nabla \cdot (\overline{V' \xi'})$, where ξ is the vorticity, and V is the horizontal wind at 200 hPa³⁸. Prime denotes an 8-day high-pass filtered wind and vorticity, and the bar represents the monthly or seasonal mean.”

Reference

Lei, T., S. Li, Luo F, N. Liu, 2020. Two dominant factors governing the decadal cooling anomalies in winter in East China during the global hiatus period. *Int. J. Climatol.*, 40(2), 750-768

REVIEWERS' COMMENTS

Reviewer #1 (Remarks to the Author):

Reviewer Summary:

In my view, the authors have addressed all my previous comments and questions, and the paper is now acceptable for Nature Communications. However, there are some grammar issues remaining throughout the text that the authors may wish to fix during the proofing stage.

Reviewer #2 (Remarks to the Author):

The authors dealt with most of my concerns reasonably. I would like to recommend an acceptance by NComm.